# Correlation between the Photographic Cranial Angles and Radiographic Cervical Spine Alignment

**DOI:** 10.3390/ijerph19106278

**Published:** 2022-05-22

**Authors:** Tomoko Kawasaki, Shunsuke Ohji, Junya Aizawa, Tomoko Sakai, Kenji Hirohata, Hironobu Kuruma, Hirohisa Koseki, Atsushi Okawa, Tetsuya Jinno

**Affiliations:** 1Department of Rehabilitation Medicine, Graduate School of Medical and Dental Sciences, Tokyo Medical and Dental University, Tokyo 1138510, Japan; petittomoko.k0215@gmail.com (T.K.); ohji.spt@tmd.ac.jp (S.O.); hirohata.spt@tmd.ac.jp (K.H.); okawa.orth@tmd.ac.jp (A.O.); jinnot@dokkyomed.ac.jp (T.J.); 2Hiro-o Orthopedics Clinic, Tokyo 1500012, Japan; hiro.orinri@gmail.com; 3Clinical Center for Sports Medicine and Sports Dentistry, Tokyo Medical and Dental University, Tokyo 1138519, Japan; j.aizawa.ue@juntendo.ac.jp; 4Division of Physical Therapy, Tokyo Metropolitan University, Tokyo 1168551, Japan; kuruma@tmu.ac.jp; 5Department of Physical Therapy, Faculty of Health Science, Juntendo University, Tokyo 1138421, Japan; 6Department of Orthopaedic Surgery, Dokkyo Medical University Saitama Medical Center, Saitama 3438555, Japan

**Keywords:** cranial vertical angle, cranial rotation angle, neck alignment, neck posture, head posture assessment, cervical alignment, spine position, vertical alignment, non-radiographic measurement, photogrammetry

## Abstract

The cranial vertical angle (CVA) and cranial rotation angle (CRA) are used in clinical settings because they can be measured on lateral photographs of the head and neck. We aimed to clarify the relationship between CVA and CRA photographic measurements and radiographic cervical spine alignment. Twenty-six healthy volunteers were recruited for this study. Lateral photographs and cervical spine radiographs were obtained in the sitting position. The CVA and CRA were measured using lateral photographs of the head and neck. The C2-7 sagittal vertical axis (SVA), cervical lordosis (C2-7), and occipito-C2 lordosis (O-C2) were measured using radiographic imaging as a standard method of evaluating cervical spine alignment. Correlations between the CVA and CRA on photographs and cervical spine alignment on radiographs were analyzed. The CVA and SVA were significantly negatively correlated (ρ = −0.51; *p* < 0.05). Significant positive correlations were found between CVA and C2-7 (ρ = 0.59; *p* < 0.01) and between CRA and O-C2 (ρ = 0.65; *p* < 0.01). Evaluating the CVA and CRA on photographs may be useful for ascertaining head and neck alignment in the mid-lower and upper parts of the sagittal plane.

## 1. Introduction

Approximately 70% of the population has neck pain at some point; this is mainly classified as non-specific neck pain with no identifiable tissue damage [1]. One of the risk factors for non-specific neck pain is malalignment of the head and neck relative to the trunk in the sagittal plane when the individual is in the sitting position [2]. Moreover, individuals with non-specific neck pain are observed to have a forward head position [3,4] and a smaller cervical lordosis angle [5] than those without neck pain. Therefore, evaluation of head and neck alignment in the sagittal plane is important for the treatment of non-specific neck pain.

The standard method of assessing head and neck alignment involves performing measurements on radiographs. These measurements can accurately ascertain the alignment in the sagittal plane based on the relative position of each joint and bone, and measurement reproducibility values are high [6]. However, radiography involves the risk of radiation exposure and requires expensive equipment and professional skills [7,8]. These are limitations to using radiographic measurements for evaluating head and neck alignment. Therefore, in a clinical setting, photographs are often used to evaluate the alignment of body parts based on the relative positions of the bony indices on the body surface [8,9].

The cranial vertical angle (CVA) and cranial rotation angle (CRA) are photographic measurements of head and neck alignment; both angles are measured on lateral photographs and have high measurement reproducibility values [10,11,12]. Theoretically, the CVA and CRA correspond to the sagittal alignment of the mid-lower and upper cervical spine, respectively. The CVA shows the inclination of the line from C7, the lowest point of the cervical spine, to the tragus of the ear, which is the dividing point between the neck and the head and is considered to reflect the flexion angle of the mid-lower cervical spine. The CRA represents the inclination of two points (the tragus of the ear and the lateral canthus of the eye) located within the head relative to the inclination of the mid-lower cervical spine and shows the inclination of the head in the sagittal plane, which is considered to reflect the flexion angle of the upper cervical spine during rotational motion. However, the relationships between the CVA and CRA on photographs and cervical spine alignment on radiographs have not been reported, and the evidence on evaluating alignment of the mid-lower or upper cervical spine using these angles remains unknown. The strength of CVA and CRA measurements is that these can be performed separately on the upper and mid-lower cervical spine, provided that a camera and some space are available. Thus, easy measurement of head and neck alignment may promote studies of neck pain.

This study aimed to clarify the relationships between CVA and CRA measurements on photographs and cervical spine alignment on radiographs. During this study, we hypothesized that the CVA and CRA were correlated with mid-lower and upper cervical alignment, respectively, on radiographs.

## 2. Materials and Methods

### 2.1. Participants

The study participants were healthy Japanese volunteers. Participants were recruited at one orthopedic clinic. Recruitment was conducted by poster advertisement. Patients were fully informed about the procedure and about the risks and social benefits of radiation exposure. Individuals were included if they met the following criteria: age 18–65 years at the time of measurement; no current neck pain; and no history of neck pain during the past 3 months. Individuals were excluded if they met the following criteria: history of trauma or fracture of the spine; inflammatory arthritis such as ankylosing spondylitis; posterior longitudinal ligament ossification; scoliosis; congenital spinal deformity; increased thoracic kyphosis; neurological injury involving the spine; rheumatoid arthritis; previous spine surgery; pain symptoms in the neck or shoulder area; and pregnancy. The sample size was calculated to be 26 subjects using G*power statistical software (version 3.0; Franz Faul, University of Kiel, Kiel, Germany) [13] with reference to the correlation coefficient between radiometric and surface measurements of spinal parameters (effect size (r) = 0.5; alpha = 0.05; power = 0.8; two-tailed) [14,15]. According to the categories of risk and corresponding levels of benefit set out in ICRP Publication 62 [16], the level of societal benefit in this study corresponds to “Intermediate”, and the corresponding effective dose range is set at 0.1–1.0 mSv. The average effective dose for radiography is reported as 0.14 mSv. The effective dose to which healthy volunteers might be exposed in this study was approximately 0.14 mSv, which is within the corresponding effective dose range. Therefore, the exposure of healthy volunteers in this study was justified. All participants provided written informed consent prior to participation. This study was approved by the university’s institutional review board (approval no. M2019-040).

### 2.2. Photographic Measurements

The heads and necks of the participants were photographed from the lateral side. Photographs were obtained using a digital video camera (HDR-CX720V; Sony Corp., Tokyo, Japan). The focal length was set at 26.0 mm to minimize photographic distortion [17]. The distance between the camera lens and the participant was 300 cm [18]. The height of the camera lens was adjusted to be at the level of the lateral canthus of the participant, and the lateral canthus was captured in the center of the image. Participants were seated with the head and trunk in the upright position and allowed to gaze forward; the height of the chair was 40 cm. The arms were extended, and the hands were placed on either side of the body. In order to standardize the photographs, the same camera was used throughout the study, the lens was always set parallel to the subject and perpendicular to the floor, and the photographs were taken using the same settings [17].

Palpation and marking were performed in the measurement position to reduce any error that might occur because of skin movement. Blue adhesive dots, 8 mm in diameter, were posted on the C7 spinous processes, the tragus of the ear, and the lateral canthus of the eye. The inferior end of the C7 spinous process was identified using the flexion-extension palpation method [19], and a marker was placed on the skin surface at that level. Using the flexion-extension palpation method, the two most prominent cervical spinous processes were palpated by the investigator’s index and middle fingers while the seated patient’s cervical spine was in flexion. Then, an assisted movement of the cervical spine was performed to place it into extension. If the upper palpated cervical spinous process moved anteriorly while the lower spinous process remained stationary, then the lower cervical spinous process was labeled C7. If both of the palpated spinous processes remained stationary, then the upper cervical spinous process was believed to be C7, and the palpation process was repeated by moving the cephalad one level at a time to confirm the level of C7 [19].

The CVA and CRA were measured using lateral photographs of the head and neck (Figure 1). The CVA was calculated by measuring the angle formed by the line connecting C7 with the tragus of the ear and the horizontal line [20]. In contrast, the CRA was determined by measuring the angle formed by the line connecting C7 with the tragus of the ear and the line connecting the tragus of the ear with the lateral canthus of the eye [20]. The intra-rater and inter-rater reliabilities of these methods were high (intraclass correlation coefficient (ICC) = 0.88–0.96) [10,11,12]. Each image was measured by a physical therapist using ImageJ (National Institutes of Health, Bethesda, MD, USA).

### 2.3. Radiographic Measurements

The lateral radiographs of the cervical spine were obtained while the participant maintained the same sitting posture immediately after the photographs were obtained [21]. The radiographs were obtained using a medical X-ray system (rotating anode DRX-1724B; Toshiba Corp., Tokyo, Japan). The radiographic film cassette was 200 cm away from the X-ray tube [21]. The imaging center was focused on the external acoustic meatus. One radiologic technologist who was trained to set proper cervical positions obtained all the radiographs.

The C2-7 sagittal vertical axis (SVA), cervical lordosis (C2-7), and occipito-C2 lordosis (O-C2) were measured using radiographic imaging as a standard assessment of cervical spine alignment (Figure 2). The SVA is the distance between a plumb line dropped from the centroid of C2 and the posterosuperior aspect of C7 [22]. This distance shows the anterior-posterior displacement of the head from the trunk, and positive values indicate anterior displacement of the head [23]. C2-7 was determined by drawing a line parallel to the inferior endplate of C2 and another line parallel to the inferior endplate of C7. Then, perpendicular lines were drawn from each of the two lines noted, thereby creating the subtended angle between the crossing of the perpendicular line [23]. This angle shows the sagittal alignment of the mid-lower cervical spine, and positive values indicate extension [24,25]. O-C2 was determined by measuring the angle between the McGregor line and the line parallel to the inferior endplate of C2 [6]. This angle shows the sagittal alignment of the upper cervical spine, and positive values indicate extension [21,24]. The intra-rater and inter-rater reliabilities of these methods were high (ICC = 0.91–0.99) [6,26,27]. Each image was measured by a physical therapist using ImageJ.

### 2.4. Statistical Analyses

All analyses were performed using non-parametric tests due to the small sample size and the deviation of certain variables from normality. Distributed data were summarized as medians and interquartile ranges. The Spearman rank correlation coefficients between the CVA and C2-7, SVA, CRA, and O-C2 were calculated. The regression lines were created by the least-squares method from the scatter plots. Each angle may have been influenced by sex and body mass index (BMI). Therefore, each angle was compared between male and female participants using the Mann–Whitney U test, and correlations between demographic data and CVA or CRA were analyzed using Spearman rank correlation coefficients. Regression analysis was performed using the forced entry method, with the parameters that were significant in the correlation analysis as the independent variables and CVA or CRA as the dependent variables. All data were analyzed using SPSS (version 21.0; IBM Corp, Armonk, NY, USA). Statistical significance was set at *p* < 0.05. 

## 3. Results

Twenty-six participants were included in the study. Of these, 65% (*n* = 17) were male and 35% (*n* = 9) were female. Participant characteristics are shown in Table 1. Descriptive data for each value are presented in Table 2. No significant differences were noted in each value when male and female participants were compared (Table 2).

The CVA and SVA had a significant negative correlation (ρ = −0.51; *p* < 0.05) (Figure 3), and the CVA and C2-7 had a significant positive correlation (ρ = 0.59; *p* < 0.01) (Figure 4). Similarly, the CRA and O-C2 showed a significant positive correlation (ρ = 0.65; *p* < 0.01) (Figure 5).

Table 3 shows the results of correlations between demographic data and CVA or CRA. There was a correlation between CVA and age (Table 3). Multiple regression analysis with CVA as the dependent variable showed a significant partial relation with C2-7 and SVA and no significant relation with age (Table 4). A simple regression analysis with CRA as the dependent variable showed a significant partial relation with O-C2 (Table 5). 

## 4. Discussion

This cross-sectional study analyzed CVA and CRA measurements on photographs and cervical spine alignment on radiographs in healthy adults and their correlations with variables showing anterior displacement of the head and sagittal alignment of the mid-lower and upper cervical spine on radiographs. The results of the study support our hypothesis. Additionally, these angles were not correlated with BMI and age. This indicates that cervical spine alignment can be measured on the body surface of individuals who are not cervically deformed with minimal effect of age and irrespective of their body shape. Moreover, this measurement could be used to evaluate cervical spine alignment without the risk of radiation exposure by radiography, in clinical rehabilitation settings, and outside of hospitals, even when frequent measurements are required.

During this study, a significant negative correlation was found between the CVA and SVA, indicating the anterior displacement of the head. To the best of our knowledge, no other study has reported this correlation. Harrison et al. [28] reported a negative correlation between the CVA and the horizontal distance between the ankle and the tragus, indicating the anterior displacement of the head in young adults without neck pain. Lau et al. [29] reported a negative correlation between the CVA and anterior head translation, indicating the anterior displacement of the head, by measuring anterior head displacement on radiographs and the CVA using an electronic angle analyzer for patients with neck pain in the standing position. Our findings were similar to those of previous studies that used other variables to show anterior displacement of the head; nevertheless, the measurement positions or the outcomes were different. The results of this, and previous, studies suggest that the decrease in the CVA on photographs indicates an increase in the anterior displacement of the head.

In contrast, our results also demonstrated a significant positive correlation between CVA and C2-7, indicating sagittal alignment of the mid-lower cervical spine. Similarly, no study has reported the correlation between the CVA and C2-7. A study of cervical spine kinematics in the sagittal plane, using radiographs of healthy adults, reported that flexion of the mid-lower cervical spine leads to a forward position of the head, while the extension of the lower cervical spine leads to a backward position of the head [30]. The C2-7 angle decreases with flexion of the mid-lower cervical spine and increases with extension [24,25]. Conversely, the CVA decreases with a forward position and increases with a backward position of the head [29]. These results support the finding that the C2-7 angle and CVA decrease with flexion of the mid-lower cervical spine and increase with the extension of the mid-lower cervical spine. Moreover, our findings suggest that the increase in the CVA on photographs may be a surrogate measure for the increase in the extension angle of the mid-lower cervical spine.

Furthermore, this study showed a significant positive correlation between the CRA and O-C2. Kinematically, the CRA and O-C2 angle increase with the extension of the upper cervical spine and decrease with flexion [21,24,31]. A study comparing head and neck sagittal alignment in healthy adults showed a significant negative correlation between the CRA on photographs and the craniovertebral angle on radiographs, thus illustrating sagittal alignment of the upper cervical spine [32]. This means that the CRA increases with extension and decreases with flexion of the upper cervical spine. Similarly, no studies have reported the correlation between the CRA and O-C2; however, our findings support the results of previous similar studies. The results of this study and previous studies suggest that the increase in the CRA on photographs shows the increase in the extension angle of the upper cervical spine.

Radiographic measurements can accurately measure cervical spine alignment in the sagittal plane based on the relative positions of each joint and bone. However, radiography requires professional skills and involves the risk of radiation exposure, which could be harmful to the body [7]. The CVA and CRA on photographs are widely used to evaluate the forward head position; however, their relationships with mid-lower cervical alignment and upper cervical alignment in the sagittal plane are still unknown [9,10,31]. The CVA and CRA are simple to use in the clinical setting because they can be measured using only a digital camera and have high intra-rater and inter-rater reliability values [10,11,12]. Our results suggest that performing CVA and CRA measurements on photographs may be a useful and simple method of evaluating the head and neck alignment of the mid-lower and upper cervical spine in the sagittal plane, respectively. Furthermore, future studies, including analyses of the flexion and extension positions, may be useful for evaluating the flexion-extension range of motion of the cervical spine on the body surface. This method allows measurements to be taken in any location and on any subject. Using these measurements, feedback on the head and neck alignment and objective comparisons before and after treatment can be made. Evaluating the head and neck alignment without the risk of radiation exposure is beneficial and may be useful for future studies on neck pain. Currently, it remains unclear whether head and neck malalignment is the cause or the result of neck pain. In the future, applying the methods described in this study for clarifying the association between malalignment as a symptom and cervical problems could be useful for the evaluation and treatment of patients.

Nevertheless, this study had some limitations. First, it is not known whether the results of this study can be applied to individuals with cervical spine disease, such as neck pain or spinal deformities. For instance, a forward head posture with severe kyphosis leads to the overall extension of the cervical spine, which may result in an opposing correlation between CVA and C2-7. Studies of individuals with a wide range of age groups and individuals with cervical spine disease are necessary to clarify these limitations. Second, C2-7 had a large variation in the descriptive data. C2-7 increases with extension and decreases with flexion of the mid-lower cervical spine. Some of the participants in this study had flexion alignment, while others had extension alignment. In addition, due to the limited number of participants, the results of this study may not be generalizable. Future research is therefore needed to confirm the generalizability of this study’s observations to different subjects, including subjects with a wide age range and with neck pain.

## 5. Conclusions

This cross-sectional study analyzed the correlations between the CVA and CRA measurements on photographs and cervical spine alignment on radiographs in healthy adults. The CVA had a significant negative correlation with SVA, demonstrating an anterior displacement of the head, and a significant positive correlation with C2-7, showing the sagittal alignment of the mid-lower cervical spine. Conversely, the CRA had a significant positive correlation with O-C2, showing the sagittal alignment of the upper cervical spine. The CVA and CRA measurements on photographs may be useful as simple tools for evaluation of the head and neck sagittal alignment of the mid-lower and upper cervical spine, respectively.

## Figures and Tables

**Figure 1 ijerph-19-06278-f001:**
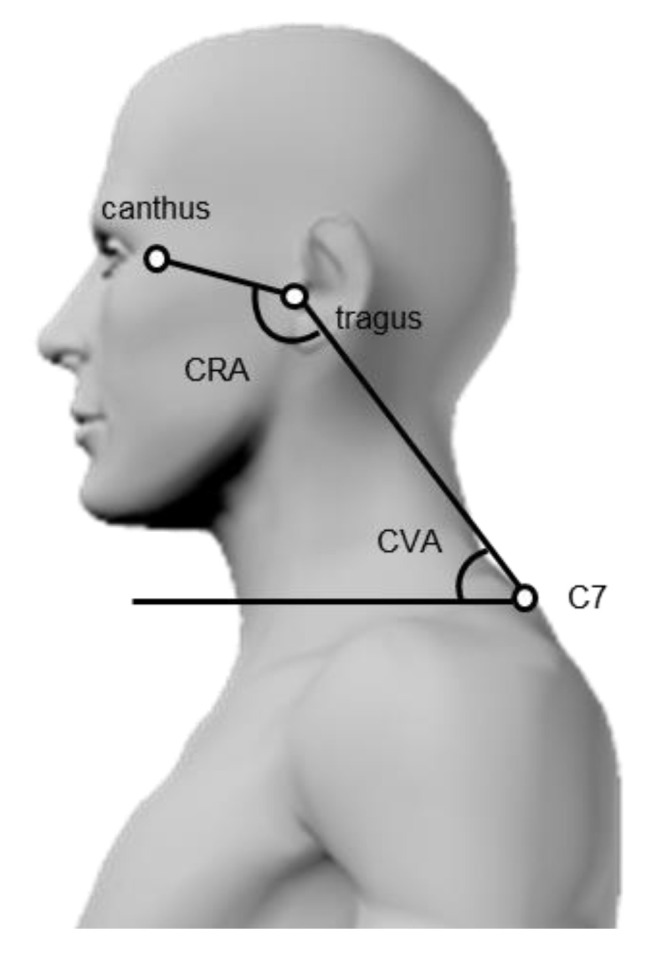
Measurement of cervical sagittal alignment on photographs. Cranial vertical angle (CVA): C7–tragus–horizontal angle; Cranial rotation angle (CRA): canthus–tragus–C7 angle.

**Figure 2 ijerph-19-06278-f002:**
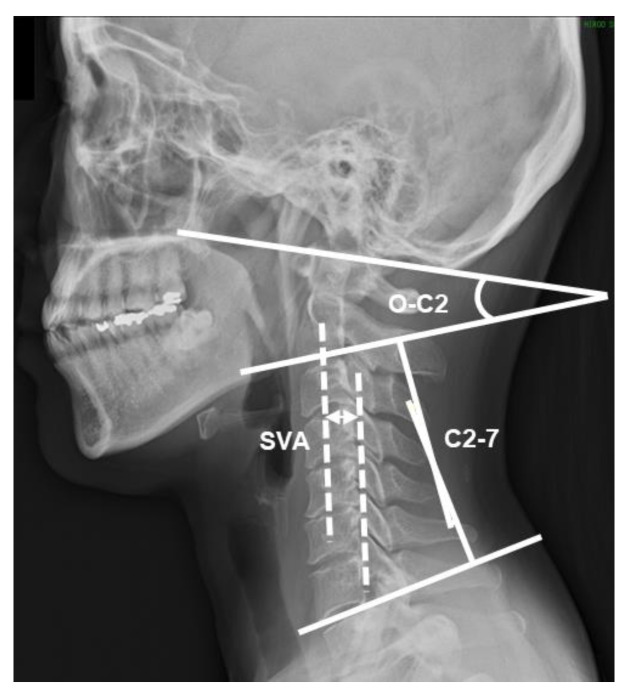
Measurement of cervical sagittal alignment on radiographs. SVA: C2-7 sagittal vertical axis; C2-7: cervical lordosis; O-C2: occipito-C2 lordosis.

**Figure 3 ijerph-19-06278-f003:**
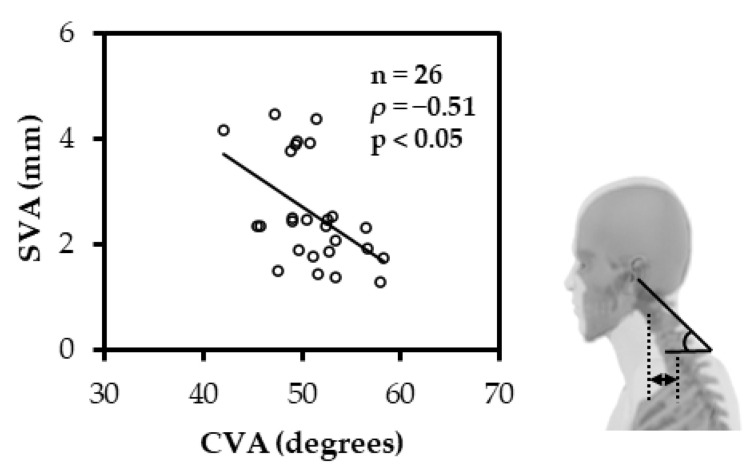
Spearman rank correlation between CVA and SVA. The CVA and SVA had a significant negative correlation. (CVA: cranial vertical angle; SVA: C2-7 sagittal vertical axis).

**Figure 4 ijerph-19-06278-f004:**
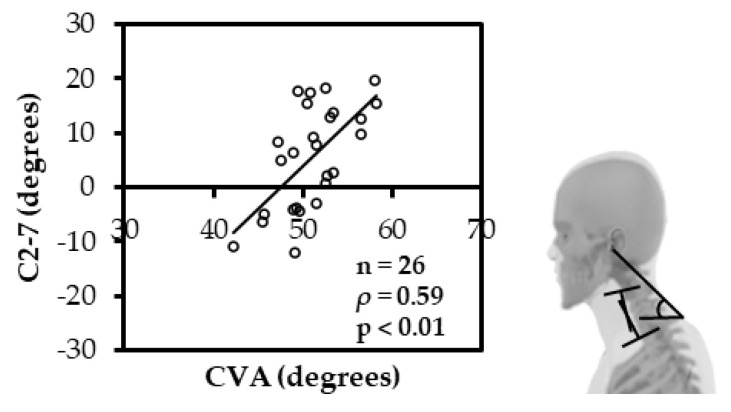
Spearman rank correlation between CVA and C2-7. The CVA and C2-7 had a significant positive correlation. (CVA: cranial vertical angle; C2-7: cervical lordosis).

**Figure 5 ijerph-19-06278-f005:**
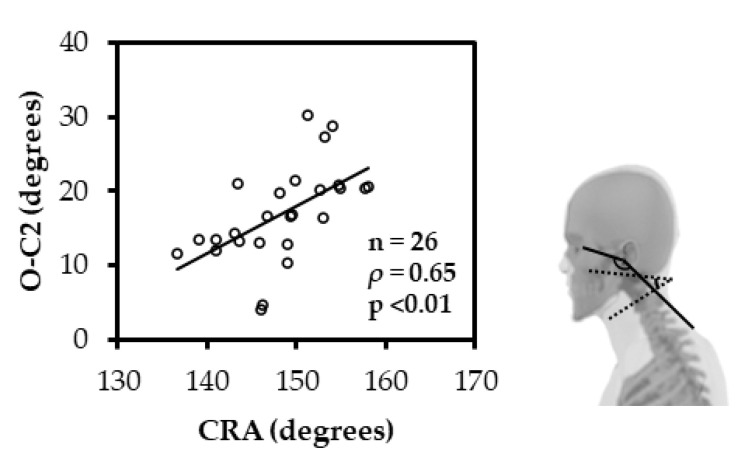
Spearman rank correlation between CRA and O-C2. The CRA and O-C2 showed a significant positive correlation. (CRA: cranial rotation angle; O-C2: occipito-C2 lordosis).

**Table 1 ijerph-19-06278-t001:** Participant characteristics (*n* = 26).

Demographic Variables		Median or Count (*n*)	IQR or %	Min	Max
Sex	Male	17	65		
	female	9	35		
Age, y (median (IQR))		21.5	2.8	18.0	36.0
Height, cm (median (IQR))		169.0	12.0	148.0	183.0
Weight, kg (median (IQR))		63.0	12.8	47.0	107.0
Body mass index, kg/m^2^ (median (IQR))		22.1	2.2	19.2	35.8

IQR: Interquartile range.

**Table 2 ijerph-19-06278-t002:** Sex differences for each value (*n* = 26).

Cervical Sagittal Alignment	Total			Male		Female		
	Median	IQR	Min/Max	Median	IQR	Median	IQR	*p* Value
CVA (degrees)	50.9	3.8	42.2/58.2	51.4	4.3	49.5	3.7	0.44
CRA (degrees)	148.9	8.7	136.6/158.1	148.9	7.4	149.3	7.8	0.81
SVA (mm)	2.3	1.9	1.3/4.5	2.1	0.7	2.5	1.4	0.06
C2-7 (degrees)	6.3	17.6	−12.1/20.1	6.3	18.1	8.3	12.3	0.57
O-C2 (degrees)	16.6	7.7	4.1/30.3	14.3	8.2	19.8	4.6	0.13

IQR: Interquartile range. CVA: cranial vertical angle; CRA: cranial rotation angle; SVA: C2-7 sagittal vertical axis; C2-7: cervical lordosis; O-C2: occipito-C2 lordosis.

**Table 3 ijerph-19-06278-t003:** Spearman rank correlation between CVA, CRA, and demographic data (*n* = 26).

		Age	Height	Weight	BMI
CVA	ρ	−0.40 *	−0.04	−0.11	−0.24
	95%CI	−0.68–−0.02	−0.42–0.35	−0.48–0.29	−0.57–0.16
CRA	ρ	0.11	−0.03	0.17	0.30
	95%CI	−0.29–0.47	−0.41–0.36	−0.23–0.52	−0.10–0.62

* *p* < 0.05. 95%CI: 95% Confidence interval. CVA: cranial vertical angle; CRA: cranial rotation angle; BMI: body mass index.

**Table 4 ijerph-19-06278-t004:** Multivariate regression for CVA as the dependent variable.

Independent Variable	Partial Regression Coefficient	Standardized Partial Regression Coefficients	*p* Value	95% Confidence Interval
Constants	57.019		0.000	48.45–65.59
SVA (mm)	−1.514	−0.392	0.019	−2.76–−0.27
C2-7 (degrees)	0.187	0.464	0.012	0.05–0.33
Age (years)	−0.137	−0.164	0.334	−0.42–0.15

R^2^ = 0.534, ANOVA F value = 8.4 (degree of freedom = 3). CVA: cranial vertical angle; SVA: C2-7 sagittal vertical axis; C2-7: cervical lordosis.

**Table 5 ijerph-19-06278-t005:** Simple regression for CRA as the dependent variable.

Independent Variable	Partial Regression Coefficient	Standardized Partial Regression Coefficients	*p* Value	95% Confidence Interval
Constants	139.867		0.000	134.4–145.4
O-C2 (degrees)	0.500	0.568	0.002	0.20–0.81

R^2^ = 0.323, ANOVA F value = 11.4 (degree of freedom = 1). CRA: cranial rotation angle; O-C2: occipito-C2 lordosis.

## Data Availability

The data that support the findings of this study are available from the corresponding author upon reasonable request. The data are not publicly available because they contain information that could compromise the privacy of the research participants.

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
