# Peer review of "Correlation between the Photographic Cranial Angles and Radiographic Cervical Spine Alignment"

_ijerph, 2022, doi:10.3390/ijerph19106278_

Round 1

Reviewer 1 Report

Kawasaki et al. report about the importance of CRA and CVA in cervical spine radiography.

Introduction:

But is the "forward head position" cause or consequence of the neck pain?

The malposition itself is a symptom, not necessarily a disease.

The immediate clinical benefit is uncertain, please specify.

Materials and Methods

Where were the individuals recruited? How has this been done?

Why did you estimate an effect size of r=0.5? What was the rationale behind that?

How was the photograph standardized?

Radiography: The healthy participants received a lateral radiograph without any clinical reason? Please name the ethics committee that accepted to x-ray healthy participants in a region with the thyroid and brain.

2.3 The first paragraph does not belong here.

Results

Please add a multivariate analysis on parameters like gender, age, and each of your angles.

Please add a standard deviation for all pearson correlations.

In the methods, you state to use a spearman correlation, in Fig.5, a pearson correlation is used. Why is that? Please be consistent.

Please give a sensitivity and specificity assessment of your photograph parameters.

Discussion

Is this test applicable to the clinical situation at all? Could photographs rule out clinical injuries at all? What is the overal benefit from the measurements you are proposing?

Also, why is this useful in healthy participants? Would it not make more sense to additionally photograph patients that already received a x-ray because of symptomatic upper neck pain?

Author Response

We thank the reviewer for the thoughtful and constructive feedback provided regarding our manuscript. We also appreciate the time and effort you have dedicated to providing insightful feedback on ways to strengthen our paper. Thus, it is with great pleasure that we resubmit our article for further consideration. We have incorporated changes that reflect the detailed suggestions you have graciously provided. Furthermore, we hope that our edits and the responses we have provided below satisfactorily address all the issues and concerns you have noted. 

Introduction

Comment 1: But is the "forward head position" cause or consequence of the neck pain?

Response: We are grateful to the Reviewer for raising this important question. Currently it remains unclear which of the two is the cause, and which is the consequence. We consider that this represents an important issue that needs to be resolved for understanding neck pain better. We also believe it is very important to establish the validity of CRA and CVA, in order to study head and neck alignment of patients with neck pain more extensively. This point has been added to the Discussion section of the revised manuscript.

Changes (Page 9, lines 343-344): Currently, it remains unclear whether head and neck malalignment is the cause or the result of neck pain.

Comment 2: The malposition itself is a symptom, not necessarily a disease.

Response: We can agree with this point. Clarification of the relationship between malposition as a symptom and neck pain could be helpful in the evaluation and treatment of patients with neck pain. This study is considered a pilot study, and we would like to conduct further research on neck pain using this method in the future. This point was added to the Discussion section of the revised manuscript.

Changes (Page 9, lines 344-346): In the future, applying the methods described in this study for clarifying the association between malalignment as a symptom and cervical problems could be useful for the evaluation and treatment of patients.

Comment 3: The immediate clinical benefit is uncertain, please specify.

Response: Thank you for your question. The immediate benefit is that alignment evaluation can be performed separately on the upper and mid-lower cervical spine, if a camera and some space are available. By measuring cervical flexion and extension and comparing these before and after treatment, it may be possible to evaluate the effects and changes on the cervical muscles and joints. This information has been added to the Introduction section of the revised manuscript.

Changes (Page 2, lines 64-67): The strength of CVA and CRA measurements is that these can be performed separately on the upper and mid-lower cervical spine, provided that a camera and some space are available. Thus, easy measurement of head and neck alignment may promote studies of neck pain.

Materials and Methods

Comment 4: Where were the individuals recruited? How has this been done?

Response: Thank you for your question. We have added the participant recruitment process to the participants sub-section, in the Materials and Methods section of the revised manuscript.

Changes (Page 2, lines 76-79): Participants were recruited at one orthopedic clinic. Recruitment was conducted by poster advertisement. Patients were fully informed about the procedure, and about the risks and social benefits of radiation exposure.

Comment 5: Why did you estimate an effect size of r=0.5? What was the rationale behind that?

Response: Thank you for your question. Cohen suggests that the effect size for the correlation analysis is large if r=0.5, medium if r=0.3, and small if r=0.1. In this study, the effect size was set to “large”, and the sample size was calculated. We have added references to this rationale in the participants sub-section, in the Materials and Methods section of the revised manuscript.

Changes (Page 2. lines 86-88): The minimum sample size calculated using G*power statistical software (version 3.0; Franz Faul, University of Kiel, Kiel, Germany) [13] was 26 individuals (effect size [r] = 0.5; alpha = 0.05; power = 0.8; two-tailed) [14, 15].

Reference (14, 15):

  1. Cohen J. (1988). Statistical Power Analysis for the Behavioral Sciences (2nd ed.). Hillsdale, NJ: Lawrence Erlbaum Associates, Publishers.
  2. Cohen J. A power primer Psychol Bull. 1992;112:155-9. DOI: 10.1037//0033-2909.112.1.155.

Comment 6: How was the photograph standardized?

Response: Thank you for your question. Consideration was given to the distance from the subject and to the focal length, in order to minimize the possibility of distortion in the image. We also maintained the same settings across all participants, in order to standardize the photograph acquisition procedure as much as possible. In the analysis software, the ratio of the image sizes were matched using a ruler projected on the image. We have added this information regarding the methods, as well as references related to photographic standardization, to the Photographic Measurements sub-section, in the Materials and Methods section of the revised manuscript.

Changes (Page 3, lines 101-102): The focal length was set at 26.0 mm to minimize photographic distortion [17].

Changes (Page 3, lines 102-104): The height of the camera lens was adjusted to be at the level of the lateral canthus of the participant, and the lateral canthus was captured in the center of the image.

Changes (Page 3, lines 107-109): In order to standardize the photographs, the same camera was used throughout the study, the lens was always set parallel to the subject and perpendicular to the floor, and the photographs were taken using the same settings [17].

Reference (17):

Nayler JR. Clinical photography: a guide for the clinician. J Postgrad Med. 2003;49:256–62.

Comment 7: Radiography: The healthy participants received a lateral radiograph without any clinical reason? Please name the ethics committee that accepted to x-ray healthy participants in a region with the thyroid and brain.

Response: Thank you for your insightful comment. A lateral radiograph was performed only in patients who consented to participate in this study, after being fully informed of the risks and social benefits of exposure to radiation. The effective dose to which healthy volunteers might be exposed in this study is approximately 0.14 mSv, which is within the corresponding effective dose range, and therefore, the exposure of healthy volunteers in this study is justified. We have added this information to the participants sub-section, in the Materials and Methods section of the revised manuscript.

In addition, the ethics committee of the "Medical Research Institute Tokyo Medical and Dental University” have approved radiography in healthy subjects. We would be grateful if you could advise us whether these details should be added to the manuscript.

Changes (Page 2, lines 88-94): According to the categories of risk and corresponding levels of benefit set out in ICRP Publication 62 [16], the level of societal benefit in this study corresponds to "Intermediate", and the corresponding effective dose range is set at 0.1-1.0 mSv. The average effective dose for radiography is reported as 0.14 mSv. The effective dose to which healthy volunteers might be exposed in this study was approximately 0.14 mSv, which is within the corresponding effective dose range. Therefore, the exposure of healthy volunteers in this study was justified.

Reference (16): UNSCEAR 2000 report - Vol. I: https://www.unscear.org/unscear/en/publications/2000_1.html

Comment 8: 2.3 The first paragraph does not belong here.

Response: Thank you for this comment. For consistency, we have also added a heading to the first paragraph of the Materials and Methods section of the revised manuscript.

Changes (Pages 2, lines 75): 2.1. Participants

Results

Comment 9: Please add a multivariate analysis on parameters like gender, age, and each of your angles.

Response: Thank you for your comment. At first, correlations between demographic data and CRA or CVA were analyzed using Spearman rank correlation coefficients. Multiple regression analysis or simple regression analysis was then performed using the forced entry method, with the parameters that were significant in the correlation analysis as the independent variables and CRA and CVA as the dependent variables. We have added the description of these methods to the Statistical Analyses sub-section in the Materials and Methods, and in the Results sections of the revised manuscript. We have also added Tables 4 and 5, and a discussion of age in the Discussion section of the revised manuscript.

Changes (Page 4, lines 173-178): Therefore, each angle was compared between male and female participants using the Mann–Whitney U test, and correlations between demographic data and CRA or CVA were analyzed using Spearman rank correlation coefficients. Regression analysis was performed using the forced entry method, with the parameters that were significant in the correlation analysis as the independent variables and CRA or CVA as the dependent variables.

Comment 10: Please add a standard deviation for all Pearson correlations.

Response: Thank you for your comment. We have added 95% confidence intervals to the correlation analysis results shown in Table 3. We hope that this revision addresses your concerns. We would be grateful if you could notify us if further changes are required.

Comment 11: In the methods, you state to use a spearman correlation, in Fig.5, a pearson correlation is used. Why is that? Please be consistent.

Response: We are grateful to the Reviewer for pointing out this issue. Fig. 5 was changed to Spearman correlation.

Comment 12: Please give a sensitivity and specificity assessment of your photograph parameters.

Response: Thank you for your comment. The parameters in this study could not be categorized as binary, and there is currently no cutoff value. We regret to say that we do not know the appropriate method for calculating sensitivity or specificity in this context. We agree, however, that a sensitivity and specificity assessment of the photograph parameters is important, and we would be grateful for your guidance on this point.

The reliability of the photographic parameters has been demonstrated in previous studies, although their validity is not clear; we believe that this study is a significant step in this direction. Nevertheless, it may be necessary to determine specific cutoff values in the future.

In this study, we referred to the reliability in previous studies and took into consideration the method of photography to maximize the reliability of the photographic measurement. We added Photographic Measurements and relevant references in the Materials and Methods section of the revised manuscript. We hope that our response is in line with the intent of your suggestion. If it is not, please do let us know.

Changes (Page 3, lines 101-109): The focal length was set at 26.0 mm to minimize photographic distortion [17]. The distance between the camera lens and the participant was 300 cm [18]. The height of the camera lens was adjusted to be at the level of the lateral canthus of the participant, and the lateral canthus was captured in the center of the image. Participants were seated with the head and trunk in the upright position and allowed to gaze forward; the height of the chair was 40 cm. The arms were extended, and the hands were placed on either side of the body. In order to standardize the photographs, the same camera was used throughout the study, the lens was always set parallel to the subject and perpendicular to the floor, and the photographs were taken using the same settings [17].

Changes (Page 3, lines 129-130): The intra-rater and inter-rater reliabilities of these methods were high (intraclass correlation coefficient [ICC] = 0.88-0.96) [10, 11, 12].

Reference (17):

Nayler JR. Clinical photography: a guide for the clinician. J Postgrad Med. 2003;49:256–62.

Discussion

Comment 13: Is this test applicable to the clinical situation at all? Could photographs rule out clinical injuries at all? What is the overal benefit from the measurements you are proposing?

Response: Thank you for your comment. This method allows measurements to be taken in any location and on any subject, provided a camera, a chair, and space of 300 cm in width, are available. However, it may need some adaptation in order to apply to all clinical situations. Although measurement accuracy may be reduced by a small extent, space with a narrower width will be enough for taking a photograph of only the required area. Although photographic measurements are not useful for diagnosing vertebral deformity or disease, they may provide information on head and neck alignment, and objective pre- and post-treatment comparisons. We believe that measurements of head and neck alignment without the risk of radiation exposure is beneficial and will be useful for future studies on neck pain. Hence, we have revised our Discussion accordingly.

Changes (Page 8, lines 339-343): This method allows measurements to be taken in any location and on any subject. Using these measurements, feedback on head and neck alignment and objective comparisons before and after treatment can be made. Evaluation of head and neck alignment without the risk of radiation exposure is beneficial and may be useful for future studies on neck pain.

Comment 14: Also, why is this useful in healthy participants? Would it not make more sense to additionally photograph patients that already received a x-ray because of symptomatic upper neck pain?

Response: Thank you for your comment. Unfortunately, this method is not directly beneficial to the healthy participants in this study. However, we believe that the social benefits are greater than the risks of exposure. We fully explained this information to all participants, and they gave their consent to participate in the study. We have added this information to the Participants sub-section, in the Materials and Methods section of the revised manuscript. It was difficult for us to take additional photographs of patients who had already had radiographs taken, considering the patients’ strain and the need to standardize settings and maintain a high accuracy of measurements.

Changes (Page 2, lines 77-79): Recruitment was conducted by poster advertisement. Patients were fully informed about the procedure, and about the risks and social benefits of radiation exposure.

Reviewer 2 Report

A very interesting and significant manuscript that represents a contribution to the assessment of the alignment of the head and neck in the sagittal plane. Very original setting of the issue, the importance of photography compared to conventional radiography, clarify the relationships between CVA and CRA measurements on photographs and cervical spine alignment on radiographs.

The introduction has been reviewed.

Method and material of the work clearly defined with the criteria for the study.

The results are presented in tabular form with the attached figures, analyzed in detail with correct statistical processing.

The conclusions are supported by the results.

References could be more recent.

Author Response

We thank you for the thoughtful and constructive feedback provided regarding our manuscript. We also appreciate the time and effort you have dedicated to providing insightful feedback on ways to strengthen our paper. Thus, it is with great pleasure that we resubmit our article for further consideration. We have incorporated changes that reflect the detailed suggestions you have graciously provided. Furthermore, we hope that our edits and the responses we have provided below satisfactorily address all the issues and concerns you have noted.

Introduction

Comment 1: The introduction has been reviewed.

Response: Thank you for your review.

Method and material

Comment 2: Method and material of the work clearly defined with the criteria for the study.

Response: We appreciate your positive review. We have added the justification for radiography, standardization of photography, and multivariate analysis, in the Materials and Methods section of the revised manuscript, according to the other Reviewers' suggestions. We would appreciate your comments on these revisions.

Changes: (Page 2, lines 88-94): According to the categories of risk and corresponding levels of benefit set out in ICRP Publication 62 [16], the level of societal benefit in this study corresponds to "Intermediate", and the corresponding effective dose range is set at 0.1-1.0 mSv. The average effective dose for radiography is reported as 0.14 mSv. The effective dose to which healthy volunteers might be exposed in this study was approximately 0.14 mSv, which is within the corresponding effective dose range. Therefore, the exposure of healthy volunteers in this study was justified.

Changes (Page 3, lines 101-109): The focal length was set at 26.0 mm to minimize photographic distortion [17]. The distance between the camera lens and the participant was 300 cm [18]. The height of the camera lens was adjusted to be at the level of the lateral canthus of the participant, and the lateral canthus was captured in the center of the image. Participants were seated with the head and trunk in the upright position and allowed to gaze forward; the height of the chair was 40 cm. The arms were extended, and the hands were placed on either side of the body. In order to standardize the photographs, the same camera was used throughout the study, the lens was always set parallel to the subject and perpendicular to the floor, and the photographs were taken using the same settings [17].

Changes (Page 4, lines 173-178): Therefore, each angle was compared between male and female participants using the Mann–Whitney U test, and correlations between demographic data and CRA or CVA were analyzed using Spearman rank correlation coefficients. Regression analysis was performed using the forced entry method, with the parameters that were significant in the correlation analysis as the independent variables and CRA or CVA as the dependent variables.

Results

Comment 3: The results are presented in tabular form with the attached figures, analyzed in detail with correct statistical processing.

Response: Thank you for your positive review. We have added multivariate analysis in the Results section and Tables 3-5 of the revised manuscript, according to the other Reviewers' suggestions. We would appreciate your comments on these revisions

Changes (Pages 7, lines 243-247): Table 3 shows the results of correlations between demographic data and CRA or CVA. There was a correlation between CVA and age (Table 3). A simple regression analysis with CRA as the dependent variable showed a significant partial relation with O-C2 (Table 4). Multiple regression analysis with CVA as the dependent variable showed a significant partial relation with C2-7 and SVA and no significant relation with age (Table 5).

Conclusions

Comment 4: The conclusions are supported by the results.

Response: We appreciate your positive review.

References

Comment 5: References could be more recent.

Response: Thank you for your comment. Reference 11 was replaced with a more recent one. We could not identify additional recent references for replacing more of the references we currently use.

Reference (11): Lee, D; Max, S; Mark, S. Internal and external sagittal craniovertebral alignment: A comparison between radiological and photogrammetric approaches in asymptomatic participants. Musculoskelet Sci Pract. 2019, 43, 12-17. DOI:10.1016/j.msksp.2019.05.003.

Reviewer 3 Report

  1. Very good design of the article, with a lot of scientific information important for practice. Correlations and interrelationships are of great importance for clinical practice!
  2. Sufficient number of authors in the references - 28.
  3. The cited authors are properly divided in the introduction and discussion. Both parts are properly structured and sufficiently descriptive. The introduction is relatively short and it would be good to expand it in order to increase the relevance of the article.
  4. The materials and methodology are described and illustrated very well. The age distribution is large (from 18 to 65). The number of patients is not large and the conclusions from the results should not be taken as general.
  5. The results are very well described, with tables and diagrams.
  6. The discussion is sufficiently voluminous and covers the information received from the authors.
  7. The conclusion is short and clear.
  8. The authors have indicated the availability of the necessary documentation for working with patients - informed consent, declaration of Helsinki and others.
  9. If the authors wish, they can cite their previous pilot studies related to the topic, which may or may not confirm their results.

Author Response

We thank you for the thoughtful and constructive feedback provided regarding our manuscript. We also appreciate the time and effort you have dedicated to providing insightful feedback on ways to strengthen our paper. Thus, it is with great pleasure that we resubmit our article for further consideration. We have incorporated changes that reflect the detailed suggestions you have graciously provided. Furthermore, we hope that our edits and the responses we have provided below satisfactorily address all the issues and concerns you have noted.

Comment 1: Very good design of the article, with a lot of scientific information important for practice. Correlations and interrelationships are of great importance for clinical practice! 

Response: We appreciate your positive review.

Comment 2: Sufficient number of authors in the references - 28. 

Response: We appreciate your positive review.

Comment 3: The cited authors are properly divided in the introduction and discussion. Both parts are properly structured and sufficiently descriptive. The introduction is relatively short and it would be good to expand it in order to increase the relevance of the article.

Response: Thank you for your constructive suggestions. We have expanded the Introduction section in the revised manuscript.

Changes (Page 2, lines 64-67): The strength of CVA and CRA measurements is that these can be performed separately on the upper and mid-lower cervical spine, provided that a camera and some space are available. Thus, easy measurement of head and neck alignment may promote studies of neck pain.

Comment 4: The materials and methodology are described and illustrated very well. The age distribution is large (from 18 to 65). The number of patients is not large and the conclusions from the results should not be taken as general.

Response: We appreciate your valuable suggestions. We have added this as a limitation in the Discussion section of the revised manuscript.

Changes (Page 9, Lines 355-359): In addition, due to the limited number of participants the results of this study may not be generalizable. Future research is therefore needed to confirm the generalizability of this study’s observations to different subjects, including subjects with a wide age range and with neck pain.

Comment 5: The results are very well described, with tables and diagrams.

Response: Thank you for your positive review. We have added multivariate analysis results and Tables 3-5 in the Results section of the revised manuscript, according to the other Reviewers' suggestions. We would appreciate your comments on these revisions.

Changes (Pages 7, lines 243-247): Table 3 shows the results of correlations between demographic data and CRA or CVA. There was a correlation between CVA and age (Table 3). A simple regression analysis with CRA as the dependent variable showed a significant partial relation with O-C2 (Table 4). Multiple regression analysis with CVA as the dependent variable showed a significant partial relation with C2-7 and SVA and no significant relation with age (Table 5).

Comment 6: The discussion is sufficiently voluminous and covers the information received from the authors.

Response: We appreciate your positive review. We have added the results of multivariate analysis to the Results section of the revised manuscript, according to the other Reviewers' suggestions. We would appreciate your comments on the revised sections.

Changes (Page 8, lines 283-286): Additionally, these angles were not correlated with BMI and age. This indicates that cervical spine alignment can be measured on the body surface of individuals who are not cervically deformed with minimal effect of age and irrespective of their body shape.

Changes (Page 9, lines 339-346): This method allows measurements to be taken in any location and on any subject. Using these measurements, feedback on head and neck alignment and objective comparisons before and after treatment can be made. Evaluation of head and neck alignment without the risk of radiation exposure is beneficial and may be useful for future studies on neck pain. Currently, it remains unclear whether head and neck malalignment is the cause or the result of neck pain. In the future, applying the methods described in this study for clarifying the association between malalignment as a symptom and cervical problems could be useful for the evaluation and treatment of patients.

Comment 7: The conclusion is short and clear.

Response: We appreciate your positive review.

Comment 8: The authors have indicated the availability of the necessary documentation for working with patients - informed consent, declaration of Helsinki and others.

Response: Thank you for your review.

Comment 9: If the authors wish, they can cite their previous pilot studies related to the topic, which may or may not confirm their results.

Response: We have no previous pilot studies. We would like to use this study as a pilot study for future research relating to neck pain.

Round 2

Reviewer 1 Report

Regarding comment 5, it is still not clear how the effect size estimation was done. The power calculation itself is straightforward, but you need to know the estimated effect size before.

Author Response

Comment: Regarding comment 5, it is still not clear how the effect size estimation was done. The power calculation itself is straightforward, but you need to know the estimated effect size before.

Response: Thank you very much for reviewing our paper again and important suggestion. There were no previous studies similar to this study. The average correlation coefficient of previous studies analyzing the correlation between photographs and radiographs on the sagittal plane of patients with adult spinal deformity is about r = 0.5. We referred to it for sample size calculation in advance. We have added prior studies as a rationale for estimating the effect size to calculate the sample size. According to this previous study, the results may differ depending on the amount of BMI, so we analyzed the correlation between each parameter and BMI in this study.

Changes (Page 2. lines 85-88): The sample size was calculated to be 26 subjects using G*power statistical software (version 3.0; Franz Faul, University of Kiel, Kiel, Germany) [13] with reference to the correlation coefficient between photographic and radiographic measurements of spinal parameters (effect size [r] = 0.5; alpha = 0.05; power = 0.8; two-tailed) [14, 15].

Reference (14):

  1. Ryan, D.J.; Stekas, N.D.; Ayres, E.W.; Moawad, M.A.; Balouch, E.; Vasquez-Montes, D.; Fischer, C.R.; Buckland, A.J.; Errico, T.J.; Protopsaltis, T.S. Clinical photographs in the assessment of adult spinal deformity: a comparison to radiographic parameters. Journal of Neurosurgery. 2021, 35, 105-109. DOI:10.3171/2020.11.SPINE201732.